

# Gene expression and immune infiltration analysis comparing lesioned and preserved subchondral bone in osteoarthritis

Gang Zhang[1,2,3,*], Jinwei Qin[4,*], Wenbo Xu[1], Meina Liu[5], Rilige Wu[6] and Yong Qin[1]

[1] The Second Affiliated Hospital of Harbin Medical University, Department of Orthopedics Surgery, Harbin Medical University, Harbin, China
[2] Department of Orthopedics, Harbin First Hospital, Harbin, China
[3] Future Medicine Laboratory, The Second Affiliated Hospital of Harbin Medical University, Harbin, China
[4] Department of Emergency, Harbin First Hospital, Harbin, China
[5] Department of Biostatistics, School of Public Health, Harbin Medical University, Harbin, China
[6] Medical Big Data Research Center, Medical Innovation Research Division of PLA General Hospital, Beijing, China
[*] These authors contributed equally to this work.

Corresponding author
Yong Qin, qinyong0125@126.com

## ABSTRACT

**Background**. Osteoarthritis (OA) is a degenerative disease requiring additional research. This study compared gene expression and immune infiltration between lesioned and preserved subchondral bone. The results were validated using multiple tissue datasets and experiments.

**Methods**. Differentially expressed genes (DEGs) between the lesioned and preserved tibial plateaus of OA patients were identified in the GSE51588 dataset. Moreover, functional annotation and protein–protein interaction (PPI) network analyses were performed on the lesioned and preserved sides to explore potential therapeutic targets in OA subchondral bones. In addition, multiple tissues were used to screen coexpressed genes, and the expression levels of identified candidate DEGs in OA were measured by quantitative real-time polymerase chain reaction. Finally, an immune infiltration analysis was conducted.

**Results**. A total of 1,010 DEGs were identified, 423 upregulated and 587 downregulated. The biological process (BP) terms enriched in the upregulated genes included "skeletal system development", "sister chromatid cohesion", and "ossification". Pathways were enriched in "Wnt signaling pathway" and "proteoglycans in cancer". The BP terms enriched in the downregulated genes included "inflammatory response", "xenobiotic metabolic process", and "positive regulation of inflammatory response". The enriched pathways included "neuroactive ligand–receptor interaction" and "AMP-activated protein kinase signaling". JUN, tumor necrosis factor $\alpha$, and interleukin-1$\beta$ were the hub genes in the PPI network. Collagen XI A1 and leucine-rich repeat-containing 15 were screened from multiple datasets and experimentally validated. Immune infiltration analyses showed fewer infiltrating adipocytes and endothelial cells in the lesioned versus preserved samples.

**Conclusion**. Our findings provide valuable information for future studies on the pathogenic mechanism of OA and potential therapeutic and diagnostic targets.

## INTRODUCTION

Osteoarthritis (OA) is a common degenerative and debilitating joint disease. It is a leading cause of disability and impaired quality of life in older adults (*Zhang et al., 2022*)[1]. OA is considered an organ disease that affects the whole joint, including cartilage, subchondral bone, meniscus, synovium, and ligaments (*Hadzic & Beier, 2023*). The pathology of OA includes cartilage loss, synovial hyperplasia, ligament fibrosis, osteophyte formation, subchondral bone remodeling and sclerosis, and increased cytokine production (*Aspden & Saunders, 2019*). These pathological alterations cause joint pain, stiffness, and dysfunction. Although articular cartilage deterioration has traditionally been regarded as the primary cause of OA and numerous cartilage-preserving treatments have been developed, emerging evidence suggests subchondral bone maintenance and remodeling also play significant roles in OA pathophysiology (*Zhu et al., 2020*). This process allows adaptation to cartilage degradation and helps maintain joint tissue homeostasis. Understanding how subchondral bone remodeling occurs in OA could provide valuable insights for the development of future therapies targeting this disease in its early stages. Subchondral bone plays a crucial role in the pathological process of OA and is an important source of pain. Many antiresorptive therapies used for osteoporosis target increased bone turnover in the subchondral region; therefore, these therapies have been explored as a means to target increased bone turnover in the subchondral region in OA, which occurs primarily in the early stages of the disease (*Eriksen et al., 2021*). *Zhu et al. (2019)* demonstrated that osteoclasts derived from subchondral bone can induce sensory innervation and osteoarthritis pain; additionally, alendronate can inhibit osteoclast activity, alleviate aberrant subchondral bone remodeling, reduce innervation, and improve pain behavior in early-stage OA. Alendronate was shown to reduce subchondral bone turnover and bone loss and preserve articular cartilage (*Zhu et al., 2013*; *Khorasani et al., 2015*). Alendronate was also found to inhibit vascular invasion into calcified cartilage in rats with OA and block osteoclast recruitment to subchondral bone and osteophytes (*Hayami et al., 2004*). However, its mechanism of action has not been elucidated.

Large quantities of data, including bioinformatics datasets, have accumulated in the "big data" era (*Pal et al., 2020*). High-throughput sequencing data collection has improved rapidly, leading to substantial advances in understanding of molecular mechanisms and the discovery of drug targets. Previous studies have demonstrated that changes in gene expression play important roles in OA diagnosis and development (*Tew et al., 2014*; *Coutinho de Almeida et al., 2019*). Significant changes in the expression of genes related to enzymatic activity and the extracellular matrix (ECM) have been observed, including those responsible for producing proteinases such as ADAMTS1, ADAMTS5, and HTRA1. These studies also revealed specific genetic components previously implicated in OA progression, such as COL1A1, HAPLN1, CILP, IL6, WNT5A, IGFBP4, APOD, and DKK3.

[1]Portions of this text were previously published as part of a preprint (*Zhang et al., 2022*).

**Table 1 Screening coexpressed DEGs using multiple datasets.**

| Datasets | Type | GPL platform | Spec | samples | Omics |
|----------|------|--------------|------|---------|-------|
| GSE51588 | Subchondral bone | GPL13497 | *Homo sapiens* | MT: LT = 20: 20 | mRNA |
| GSE30322 | Subchondral bone | GPL7294 | *Rattus norvegicus* | E-group: S-group = 15: 15 | mRNA |
| GSE114007 | Cartilage | GPL11154, GPL18573 | *Homo sapiens* | OA: Normal = 20: 18 | mRNA |
| GSE110606 | Chondrocyte | GPL18573 | *Homo sapiens* | HC: HC OA = 1: 1 | mRNA |
| GSE55457 | Synovium | GPL96 | *Homo sapiens* | OA: Healthy = 10: 10 | mRNA |
| GSE55235 | Synovium | GPL96 | *Homo Sapiens* | OA: Healthy = 10: 10 | mRNA |

An integrated approach was used to analyze miRNA and mRNA sequencing data from OA cartilage, revealing the interactome of OA miRNAs and associated pathways. The study emphasized the impact of miRNAs on gene expression in cartilage and highlighted the challenges in functionally validating a network of genes targeted by multiple miRNAs. However, the pathological changes in OA differ between the lesioned and preserved sides in terms of cartilage and subchondral bone changes (*Omoumi et al., 2015*). More research is needed to develop local therapies and precision medicine approaches for treating OA.

In this study, we compared the gene expression profiles of lesioned and preserved tibias in OA patients and identified associated genes, pathways, and immune cells. The results were validated using multiple tissue datasets and experiments.

## MATERIALS & METHODS

### Data processing and identification of DEGs

The NCBI Gene Expression Omnibus (GEO) is a public data repository that stores gene expression profiles, raw series, and platform records (https://www.ncbi.nlm.nih.Gov/geo/). The subchondral bone dataset, GSE51588, included 10 normal and 40 OA samples. The platform used was GPL13497. The number of datasets for subchondral bone samples was limited. Only GSE51588 was available for human samples, and we chose OA-Normal (10:40) for validation. The 40 OA samples included medial tibia samples (MT, significant degeneration, lesioned) and lateral tibia samples (LT, minimal degeneration, preserved). The second dataset for subchondral bone samples in GEO datasets was GSE30322, but it consisted of Sprague Dawley (SD) rats for OA-Normal (15:15). Additionally, GSE110606 contained human chondrocytes for OA-Normal (1:1), while GSE114007 included human cartilage samples for OA-Normal (20:18). Lastly, GSE55457 and GSE55457 comprised synovial tissue samples, including OA, Normal, and rheumatoid arthritis subgroups. We chose OA-Normal samples from each dataset for validation (10:10). The OA samples were used for further training analyses (Table 1). The data matrix series were downloaded, and the limma package in R software was used to identify differentially expressed genes (DEGs) (*Ritchie et al., 2015*). $P < 0.05$ and $|logFC2| > 1$ were the cutoff criteria.

## Functional annotation of DEGs

The Database for Annotation, Visualization, and Integrated Discovery (DAVID; https://david.ncifcrf.gov/) provides a comprehensive set of functional annotation tools and helps investigators understand the biological meaning behind large lists of genes. Gene Ontology (GO) annotation covers gene cell components (CCs), molecular functions (MFs), and biological processes (BPs). Kyoto Encyclopedia of Genes and Genomes (KEGG) pathway enrichment analysis extracts pathway information from molecular interaction networks. In the present study, the DAVID online tool was used for GO and KEGG pathway enrichment analyses of the upregulated and downregulated genes. R language was used for data visualization, and $P < 0.05$ was considered to indicate significant GO and KEGG terms.

## Analysis of the protein–protein interaction network and hub genes

The construction of protein-protein interaction (PPI) networks is critical for understanding cell biology and interpreting genomic data. The Search Tool for the Retrieval of Interacting Genes/Proteins (STRING; http://www.string-db.org/) is an online biological database and website designed to construct PPI networks in molecular biology. A PPI network was constructed using the STRING database to understand the DEGs more in-depth. The DEGs (species: *Homo sapiens*) were mapped to the STRING database (PPI score > 0.4), and those with a combined score > 0.99 were identified *via* Cytoscape software 3.7.1 (http://www.cytoscape.org/) to visualize the PPI network. The top 10 hub genes were identified *via* the maximal clique centrality (MCC) method.

## Screening coexpressed DEGs using multiple datasets

This study was approved by the Ethics Committee of the Second Affiliated Hospital of Harbin Medical University (approval number: ky2020-078), and we obtained written informed consent from the participants. Six GSE datasets containing data from the joint tissues of OA patients were used for validation to screen coexpressed genes in multiple tissues. The DEGs in the associated datasets, including GSE51588 (OA-Normal for subchondral bone samples), GSE30322 (subchondral bones of SD rat OA models), GSE110606 and GSE114007 (chondrocyte and cartilage samples), and GSE55235 and GSE55457 (synovial tissues), were used in this study to screen for coexpressed genes that aligned with the expression profile of the OA subchondral bone samples (Table 1). Differentially expressed genes ($|logFC2|>1$, $P < 0.05$) were considered for further validation. A Venn diagram was constructed to display the results.

## Quantitative reverse-transcription polymerase chain reaction

The study protocol was approved by the Ethics Committee of the Second Affiliated Hospital of Harbin Medical University (approval number: ky2020-078), and all of the patients provided informed consent. Total knee placement samples were collected. Ethical approval and consent for the use of resected tissue were obtained. Subchondral bone samples were obtained and stored in liquid nitrogen until analysis. Cartilage and synovium samples were cut into small pieces ($\sim 1$ mm$^3$) and transferred into 0.2% collagenase II and
**Table 2 Gene primers.**

| Genes | Forward (5′–3′) | Reverse (5′–3′) |
|---|---|---|
| LRRC15 | GCCTTTGGACAAGGCTATGC | GAGCAGGTACACTCGCTAGG |
| COL11A1 | TGGTGATCAGAATCAGAAGTTCG | AGGAGAGTTGAGAATTGGGAATC |
| GAPDH | CACTCAGACCCCCACCACAC | GATACATGACAAGGTGCGGCT |

0.4% collagenase I solutions, respectively. The samples were then incubated for 4 h at 37 °C. Then, $\alpha$-MEM supplemented with 10% fetal bovine serum and 1% penicillin–streptomycin was added, and the samples were incubated at 37 °C with 5% $CO_2$ until they reached >80% confluence. We discarded the cell debris three days later and passaged the cells three times to eradicate residual contaminating macrophages. The resulting monocultures of synovial fibroblasts (SFs) were used for experiments until passage 8 (*Pérez-García et al., 2016*). Passage 2 chondrocytes were used for experiments. SF and chondrocytes were cultured in 6-well plates, and 10 ng/ml IL-1$\beta$ was added to each well for 48 h for further analyses.

Total RNA from SF, chondrocytes, and subchondral bone was obtained using a reagent (ES Science Technique, Shanghai, China). RNA was reverse-transcribed using the ES Science cDNA Reverse Transcription Kit (ES Science Technique, Shanghai, China), and quantitative reverse-transcription polymerase chain reaction (qRT-PCR) was performed with the ES Science SYBR Green Kit (ES Science Technique, Shanghai, China). The primers used are listed in Table 2. Per the manufacturer's instructions, 1 μg of total RNA was added to 2 μL of DNA enzyme and diluted with ddH$_2$O to a final volume of 16 μL. The mixture was incubated at 25 °C for 5 min to remove genomic DNA. Then, 4 μL of 5 × RT mix (10 U/μL) was added to 16 μL of total RNA treated with DNA enzyme for reverse-transcription at 42 °C for 15 min. A qPCR system (20 μL) was prepared as follows: 10 μL of 2x Super SYBR Green qPCR Master Mix (0.05 U/μL), 2 μL of template, 0.4 μL each of forward (10 μmol/L) and reverse (10 μmol/L) primer, and ddH$_2$O to obtain a final volume of 7.2 μL. qRT–PCR was performed as follows: one cycle of 95 °C for 5 min, 40 cycles of 95 °C for 10 s, and 40 cycles of 60 °C for 30 s. For relative quantification, we normalized the target gene expression to that of the housekeeping gene (GAPDH). The results are presented as the relative expression with respect to the untreated condition using the formula $2^{-\Delta\Delta Ct}$. The experiment was repeated at least three times. The data were analyzed with GraphPad Prism 7.0 (GraphPad Software, Boston, MA, USA). Differences were assessed using Student's two-tailed $t$ test, and $P < 0.05$ was considered to indicate significance.

### Computational analysis of immune infiltrating cells

xCell is a novel and robust method based on single sample gene set enrichment analysis (ssGSEA) that estimates the abundance scores of 64 cell types in the microenvironment. There are two approaches to running xCell: one is the online website tool (https://xCell.ucsf.edu/), and the other is the xCell package in R software. We estimated the abundance scores of 64 cell types in lesioned and preserved subchondral bone samples using the online tool. $P < 0.05$ was considered to indicate significance. The immune cell scores were also calculated for each sample. The relationships among immune cells were calculated using the Pearson coefficient. The "ggplot2" package was used to construct violin

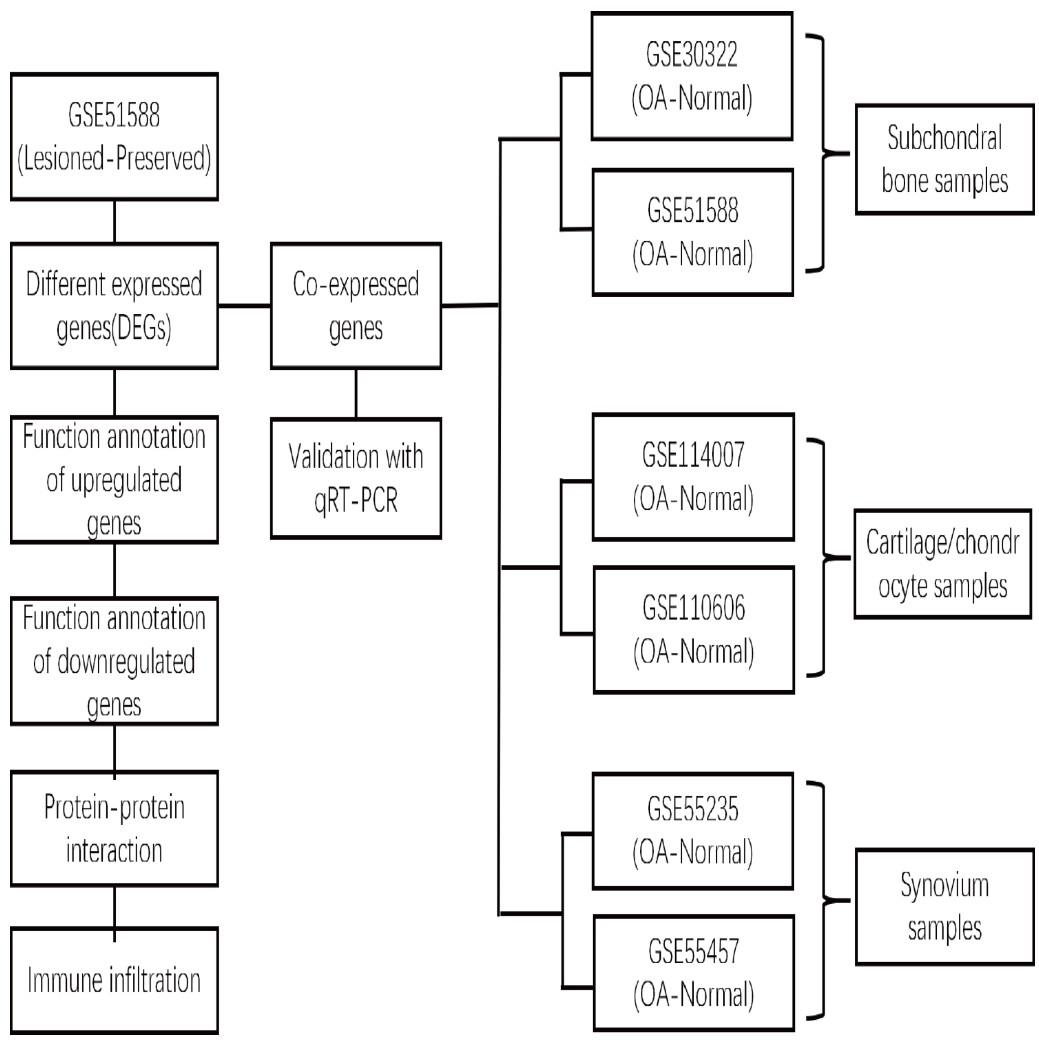

**Figure 1 Flow chart depicting the OA gene expression analysis study process.** The DEGs between the lesioned and preserved tibial plateaus of OA patients in the GSE51588 dataset were identified. Functional annotation and PPI network analysis were conducted on the lesioned and preserved sides to explore potential therapeutic targets in OA subchondral bones. Multiple tissues were used to screen for coexpressed genes in OA, and the expression levels of the identified candidate DEGs were measured by qRT–PCR. An immune infiltration analysis was also conducted.

diagrams to visualize the differences in immune cell infiltration. A flow chart depicting the study process for the OA gene expression analysis is shown in Fig. 1.

## RESULTS

### Analysis of differential gene expression in all datasets

Differential gene expression analysis was performed on OA and normal subchondral bone samples. There were 1,010 DEGs in the GSE51588 MT-LT cohort, 423 upregulated and 587 downregulated. The most upregulated genes were STMN2 (logFC = 5.52924, $P < 0.05$) and POSTN (logFC = 3.984087, $P < 0.05$), and the most downregulated genes were LEP
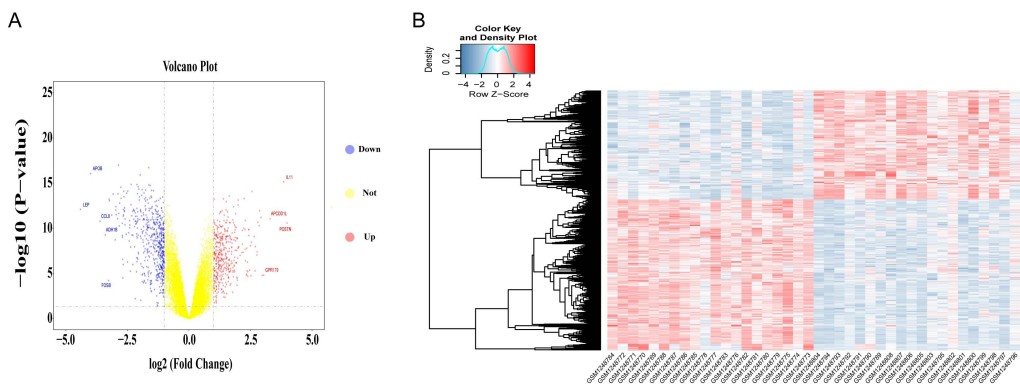

**Figure 2 Volcano plot and heatmap of DEGs between lesioned and preserved subchondral bone tissues.** (A) Volcano plots displaying the DEG results, where the $x$- and $y$-axes represent the log-transformed threshold values. Red dots indicate all significantly upregulated genes, and blue dots indicate all significantly downregulated genes that passed the screening threshold. Yellow dots represent genes that did not demonstrate a significant change. (B) DEGs selected by volcano plot filtering ($|logFC| > 1$ and $P < 0.05$).

(logFC $= -4.41131$, $P < 0.05$) and APOB (logFC $= -4.00156$, $P < 0.05$). The distribution of all DEGs according to the two dimensions of $-log10$ ($P$ value) and logFC is represented by a volcano map (Fig. 2A). The DEGs were evaluated using a heatmap (Fig. 2B). The detailed logFC and $P$ values of the DEGs are shown in the supplementary data.

## DEG functional and pathway analyses

GO annotation and KEGG pathway enrichment analyses were performed to better understand the functional significance of the DEGs. Functional enrichment analysis of the upregulated DEGs revealed that the top five biological processes were skeletal system development, sister chromatid cohesion, ossification, mitotic nuclear division, and extracellular matrix organization (Fig. 3A). The main cellular components and molecular functions involved included spindle microtubules, proteinaceous extracellular matrix, platelet-derived growth factor binding, and metalloendopeptidase activity (Figs. 3B–3C). KEGG pathway analysis of the upregulated DEGs revealed that the top five terms were associated with the Wnt signaling pathway, proteoglycans in cancer, protein digestion and absorption, the PI3K-Akt signaling pathway, and the Hippo signaling pathway (Fig. 3D).

The functional enrichment analysis of the downregulated DEGs revealed that the top five BP terms were associated with xenobiotic metabolic process, triglyceride biosynthetic process, positive regulation of inflammatory response, positive regulation of fat cell differentiation, and positive regulation of B-cell activation (Fig. 4A). The main cellular components and molecular functions were associated with the receptor complex, plasma membrane, transporter activity, and retinol dehydrogenase activity (Figs. 4B–4C). KEGG pathway analysis of the downregulated DEGs revealed that the top terms were associated with tyrosine metabolism, regulation of lipolysis in adipocytes, the PPAR signaling pathway, neuroactive ligand–receptor interaction, and the AMPK signaling pathway (Fig. 4D).

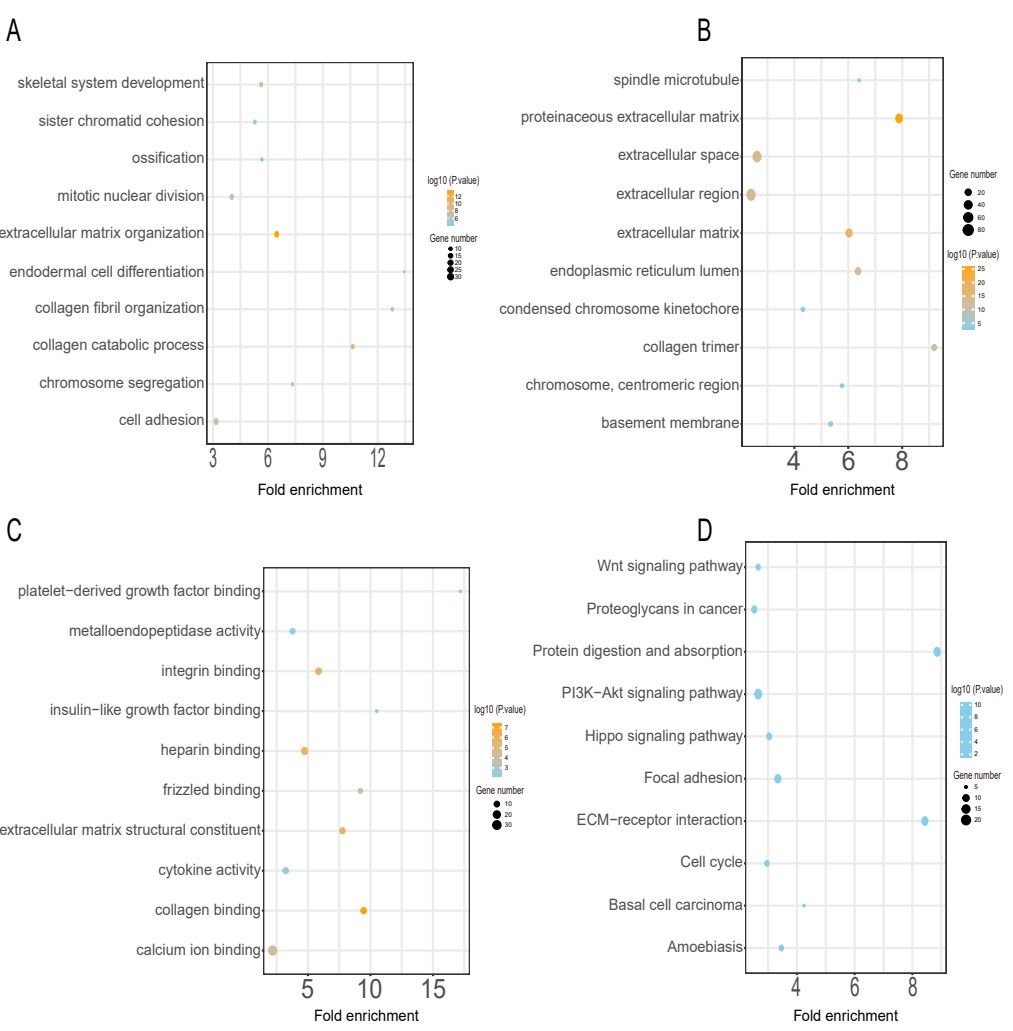

**Figure 3  GO and KEGG pathway enrichment analyses of upregulated genes.** GO functional classification of DEGs. The $x$-axis represents the number of DEGs, with individual GO terms plotted on the $y$-axis. The graph displays only significantly enriched GO terms ($P < 0.05$). All GO terms were grouped into three categories: (A) BP analysis, (B) CC analysis, and (C) MF analysis. (D) KEGG pathway analysis. The $x$-axis represents the number of DEGs annotated in a pathway, with individual KEGG terms shown on the $y$-axis. The graph displays only significantly enriched KEGG terms ($P < 0.05$).

## PPI network construction and hub gene identification

Using the STRING database, we constructed a putative PPI network map for the relationships between the DEGs, which was visualized with Cytoscape. Finally, a total of 1,010 DEGs were mapped to 5,766 nodes in the network with a combined score >0.4. Network interactions with a score >0.99, including 116 nodes, were visualized with Cytoscape software (Fig. 5A). The top 10 hub genes identified by the MCC included JUN, TNF, IL-1$\beta$, LEB, CXCL8, and FN1 (Fig. 5B).

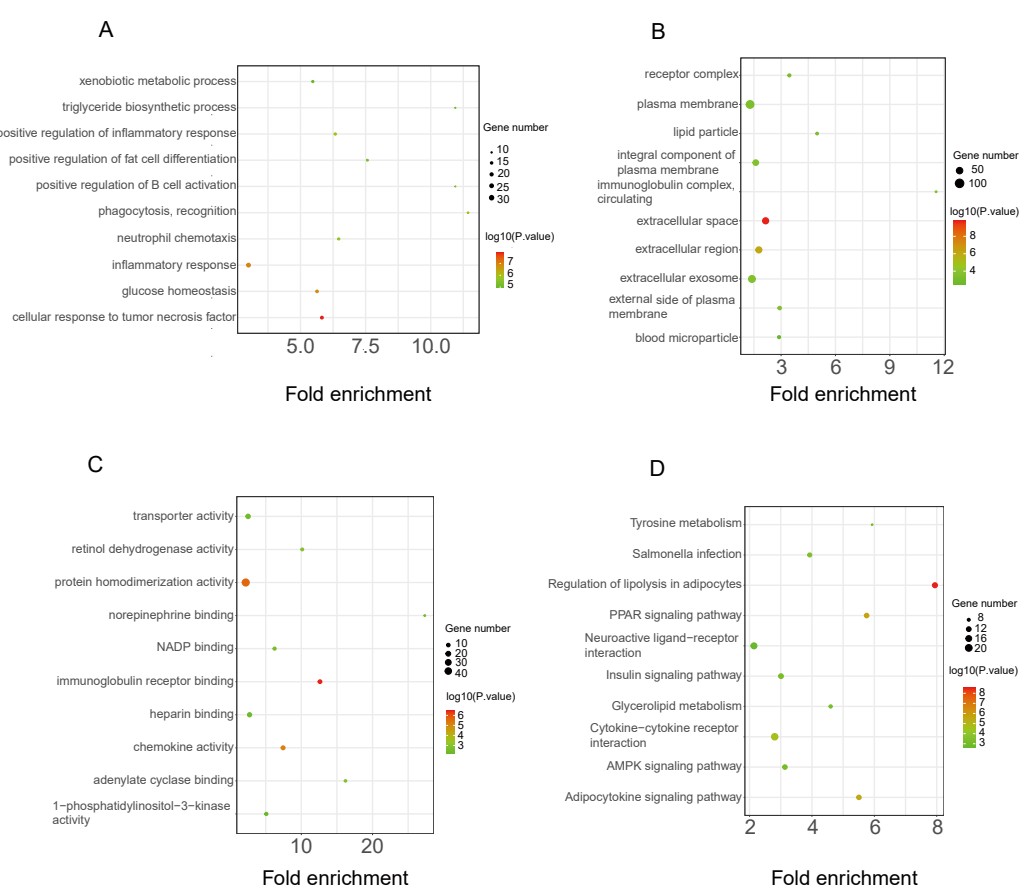

**Figure 4 GO and KEGG pathway enrichment analyses of downregulated genes.** GO functional classi-fication of DEGs. The *x*-axis represents the number of DEGs, with individual GO terms plotted on the *y*-axis. The graph displays only significantly enriched GO terms ($P < 0.05$). All GO terms were grouped into three categories: (A) BP analysis, (B) CC analysis, and (C) MF analysis. (D) KEGG pathway analysis. (D) KEGG pathway analysis. The *x*-axis represents the number of DEGs annotated in a pathway, with individual KEGG terms shown on the *y*-axis. The graph displays only significantly enriched KEGG terms ($P < 0.05$).

## Identification of coexpressed genes in datasets from multiple tissues and validation with qRT–PCR

Six multiple-tissue datasets, including subchondral bone, cartilage, and synovium samples, were used for validation. Two coexpressed genes, LRRC15 and Col11A1, were screened out. A Venn diagram was constructed to visualize the results (Fig. 6). The LRRC15 and Col11A1 genes were upregulated in all datasets. To confirm the results of our bioinformatics analysis, we conducted qRT–PCR to validate the expression of the genes coexpressed with chondrocyte, fibroblast-like synovial (FLS), and subchondral bone samples. LRRC15 and Col11A1 expression levels were higher in the OA group than in the normal group (Figs. 7A–7C).

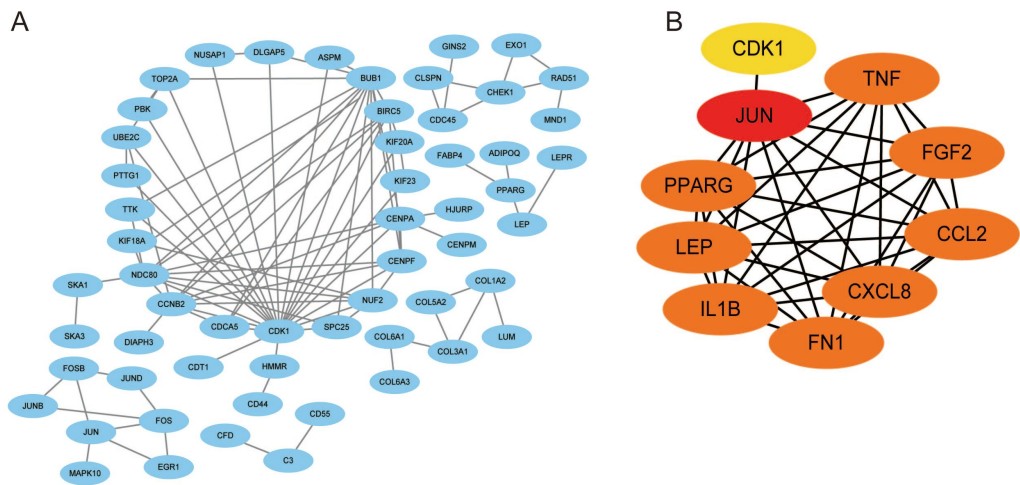

**Figure 5** **A PPI network map was used to screen for the DEGs in lesioned and preserved subchondral bone tissues.** (A) Network interactions with a score > 0.99, including 116 nodes, were visualized with Cytoscape software. (B) The top 10 hub genes identified by the MCC included JUN, TNF, IL-1β, LEB, CXCL8, and FN1.

## Analysis of immune infiltrating cells

The immune response in subchondral bones is a significant factor in OA development. Multiple immune cells participate in the maturation and activation of osteoblasts through highly complex mechanisms (*Luo et al., 2023*). We analyzed immune infiltrating cells using the xCell algorithm to assess differential immune cell infiltration between OA and normal tissues. A correlation heatmap of the 64 types of immune cells revealed that ly endothelial cells, mv endothelial cells, and endothelial cells were significantly positively correlated. CD4+ T cells and CD4+ Tem cells also demonstrated a positive correlation. (Fig. S1A). A heatmap was constructed of the abundance scores of the immune cells in each sample (Fig. S1B).

A violin plot of the differential immune cell infiltration levels revealed that fewer adipocytes and endothelial cells infiltrated in the lesioned samples *versus* the preserved samples, whereas MSCs, myocytes, plasma cells, Th2 cells, ly endothelial cells, and mv endothelial cells infiltrated more (Figs. 8A–8H).

## DISCUSSION

Despite our understanding of known risk factors, such as obesity, mechanical abnormalities, genetic predisposition, and age, the exact cause of OA remains unclear (*Deveza, Zankl & Hunter, 2023*). Thus, current therapeutic approaches for OA primarily focus on symptom relief rather than specific treatment (*Makarczyk et al., 2021*). Previous studies on OA have focused mainly on the synovium, articular cartilage, and meniscus while neglecting the role of subchondral bone in OA development (*Weber, Chan & Wen, 2019*; *Chen et al., 2020*). However, recent evidence suggests that abnormal metabolism of subchondral bone is associated with articular cartilage degeneration (*Chen et al., 2017a*). Furthermore, an

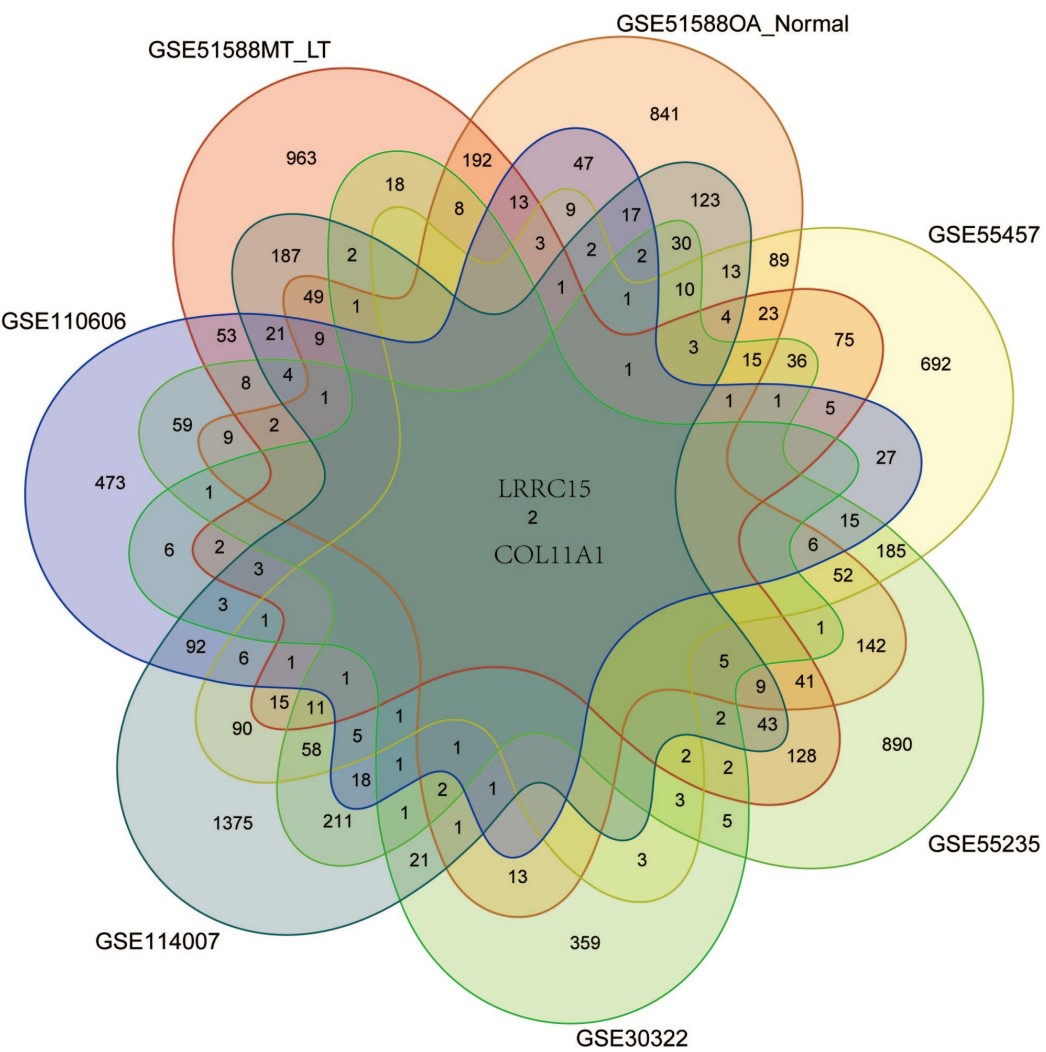

**Figure 6  Screening for coexpressed genes in multiple tissues revealed LRRC15 and Col11A1.** The Venn diagram represents the intersections of coexpressed genes across different datasets. The six datasets included cartilage, subchondral bone, and synovial tissue. The species are human and rat. Different colors and numbers represent the number of coexpressed genes across different datasets.

increasing number of research studies have indicated that the immune system plays a crucial role in the development of OA (*Fahy et al., 2015*; *Kalaitzoglou, Griffin & Humphrey, 2017*). However, the connection between immune cells and subchondral bone disturbance remains ambiguous. Bioinformatics, which combines molecular biology and information technology, holds immense importance in uncovering the molecular mechanisms underlying various diseases. This study used bioinformatics to further understand the differential gene expression and immune infiltration profiles between lesioned and preserved subchondral bone in OA. We identified dysregulated genes associated with OA progression in lesioned and preserved subchondral bone in OA patients. STMN2 and POSTN were the most upregulated genes, and LEP and APOB were the most downregulated.

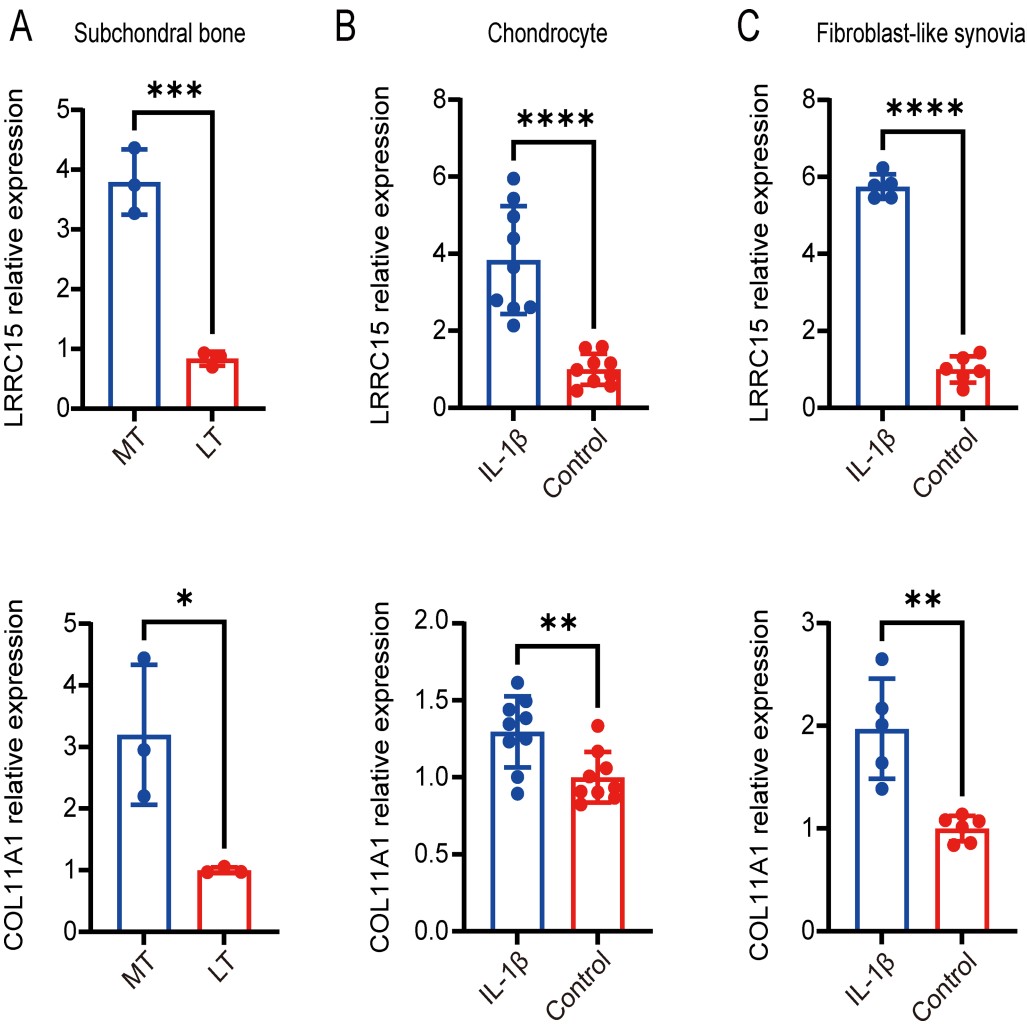

**Figure 7 Validation of coexpressed genes in multiple tissues *via* qRT–PCR.** Relative expression of LRRC15 and COL11A1 in subchondral bone (A), chondrocyte (B), and fibroblast-like synovial (C) tissue samples, as determined by qRT–PCR. Each column represents the mean ± SD. The OA group exhibited higher expression levels of LRRC15 and COL11A1 than did the normal group. MT represents remedial tibia samples (significant degeneration, lesioned), and LT represents lateral tibia samples (minimal degeneration, preserved). IL-1$\beta$ (10 ng/ml) was added to SF and chondrocytes for 48 h for further analysis. *$P$ < 0.05 *vs.* the normal group; **$P$ < 0.01 *vs.* the normal group; ***$P$ < 0.001 *vs.* the normal group; ****$P$ < 0.0001 *vs.* the normal group; $n = 3$. The experiment was repeated at least three times.

The most enriched pathway associated with the upregulated genes was the Wnt signaling pathway, while the most enriched pathway associated with the downregulated genes was tyrosine metabolism. JUN, TNF, IL-1$\beta$, and LEB were identified as hub genes. LRRC15 and Col11A1 were upregulated and coexpressed in multiple OA tissues. In addition, immune infiltration analysis revealed that many immune cells demonstrated different infiltration abundance scores.

Previous studies have focused on articular cartilage degeneration and overlooked the role of the subchondral bone and synovium. In recent years, increasing studies have

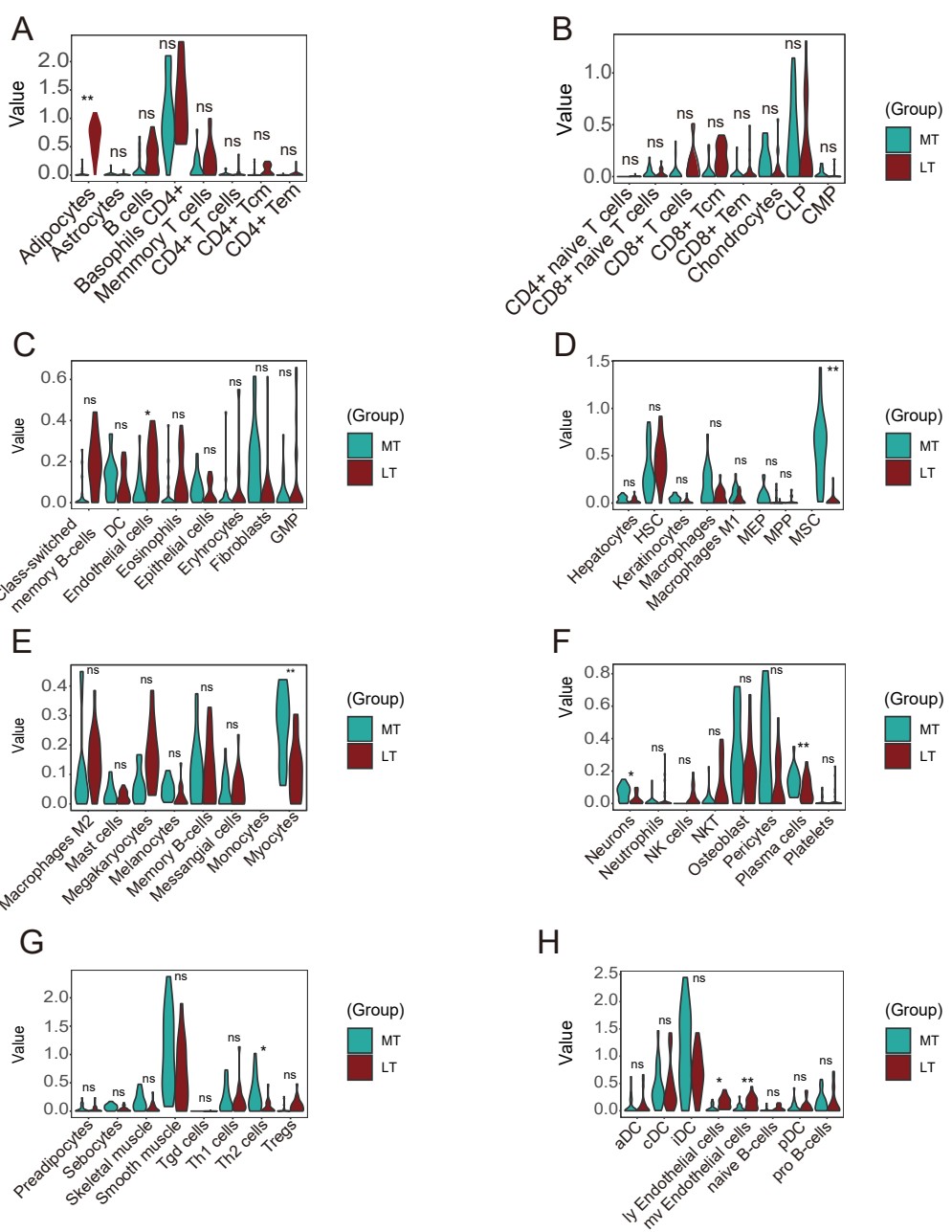

**Figure 8** **Differences in immune infiltration between lesioned and preserved subchondral bone tissues.** (A–H) Differences in immune infiltration between lesioned and preserved subchondral bone tissues. MT represents remedial tibia samples (significant degeneration, lesioned), which are marked in green. LT represents lateral tibia samples (with minimal degeneration, preserved), which are marked in brown. *P* values < 0.05 were considered to indicate significance.

demonstrated that the knee joint is an organ and that OA can affect subchondral bone, the synovium, and other tissues (*Aspden & Saunders, 2019*; *Mao et al., 2021*; *Du et al., 2023*). Subchondral bone links the joint to the diaphyseal bone, provides mechanical

support, provides some nutrition, and removes metabolic waste products (*Schilling, 2017*). Subchondral bone remodeling begins in the early stages of cartilage degeneration. The earliest signs of OA on MRI scanning in the subchondral bone are bone marrow lesions (BMLs; excessive water signals in bone). Subchondral bone remodeling is thought to arise due to mechanical overload (*Donell, 2019*). The pathological changes in subchondral bone during OA include angiogenesis, *de novo* bone formation, sensory innervation invasion, bone cysts, sclerosis, and osteophyte formation (*Wang et al., 2017a*; *Wang et al., 2017b*; *Zhu et al., 2019*; *Zhou et al., 2020*). Evidence suggests that abnormal bone remodeling may contribute to the development of OA and could be a target for OA therapy (*Chen et al., 2017a*; *Hu et al., 2021*; *Kon et al., 2021*).

The most upregulated genes were stathmin2 (STMN2) and periostin (POSTN), and the most downregulated genes were leptin (LEP) and APOB. STMN2 is associated with sensory neuron growth and contributes to regenerating axons after nerve injury (*Dubový et al., 2018*). STMN2 may also be involved in OA-related pain. POSTN is a 90-kDa member of the fasciclin family that can induce the expression of proinflammatory factors, such as MMP-9, MMP-10, and MMP-13, leading to extracellular matrix degeneration (*Ohta et al., 2014*). LEP is a peptide hormone containing 167 amino acids (*Yang et al., 2019*) that promotes the differentiation of osteoblasts under normal conditions (*Xu et al., 2016*). LEP expression was increased in OA subchondral osteoblasts and partially elevated the expression levels of alkaline phosphatase, osteocalcin release, collagen1, and TGF-$\beta$1. Given these contradictory findings, the disturbance of osteoblast, osteoclast, and subchondral bone remodeling under OA conditions (*Mutabaruka et al., 2010*) requires further study.

Among the upregulated DEGs, the most enriched pathways were the Wnt and PI3K-Akt signaling pathways. The Wnt signaling pathway can directly affect subchondral bone cartilage and synovial tissue and has been demonstrated to play important roles in pathology (*Zhou et al., 2017*; *Wang et al., 2019*). AKT phosphorylation in subchondral bone can promote osteogenic differentiation and osteoblastic proliferation, resulting in aberrant bone formation. In addition, targeting these pathways may alleviate OA development. In contrast, PI3K/AKT inhibition reduces subchondral bone sclerosis by decreasing osteogenesis (*Lin et al., 2018*). Suppressing the PI3K-Akt signaling pathway can also enhance autophagy and reduce inflammation in chondrocytes (*Xue et al., 2017*). Pharmaceutically targeting this pathway is a promising approach for OA treatment (*Sun et al., 2020*).

Regarding the downregulated DEGs, the most enriched pathways included tyrosine metabolism and AMP-activated protein kinase (AMPK) signaling. AMPK mainly regulates energy balance and metabolism. AMPK dysregulation is associated with multiple age-related diseases, including atherosclerosis, cardiovascular disease, diabetes, cancer, neurodegenerative diseases, and OA (*Jeon, 2016*; *Wang et al., 2020*). Upregulation of phosphorylated and total AMPK expression in articular cartilage can limit OA development and progression in OA animal models (*Zhou et al., 2017a*; *Zhou et al., 2017b*; *Li et al., 2020*).

Ten hub genes were identified in the PPI network, and the proteins encoded by these genes, including JUN, TNF, IL-1$\beta$, LEB, CXCL8, and FN1, were the key nodes in the PPI network. These proteins are associated with OA progression, especially JUN, IL-1$\beta$, and

TNF (*Rhee et al., 2017*; *Zhao et al., 2019*; *Ahmad et al., 2020*). JUN is a major component of the transcription factor activator protein-1 (AP-1) family (*Atsaves et al., 2019*) and can mediate catabolic transcription and cell apoptosis/death. Jun also plays an important role in the TNF and IL-17 signaling pathways (*Cai et al., 2020*).

LRRC15 and COL11A1 were identified as coexpressed genes in subsequent multiple-tissue dataset analyses and experiments. We propose that these two genes, newly associated with OA, play important roles in the disease pathogenesis. LRRC15, a type I membrane protein comprising 581 amino acids with no obvious intracellular signaling domains, is upregulated by the proinflammatory cytokine TGF$\beta$ in cancer-associated fibroblasts (*Ben-Ami et al., 2020*). LRRC15 is highly expressed on stromal fibroblasts and tumor cells, such as those in melanoma, sarcoma, and glioblastoma (*Purcell et al., 2018*). ABBV-085 is an antibody drug that targets LRRC15 and has been widely studied in antitumor research (*Hingorani et al., 2021*). Therefore, ABBV-085 may be effective for treating OA. Elevated LRRC15 expression in early OA may indicate chondrocyte activation, promoting disease progression and leading to permanent changes in the phenotype and response of OA chondrocytes. This effect can facilitate the transformation of chondrocyte function and phenotype by regulating the stress response and imbalanced remodeling of the extracellular matrix. Further research is necessary to gain a deeper understanding of how LRRC15 contributes to cartilage damage regulation and the signaling pathways related to chondrocyte function (*Singh et al., 2021*). In addition, LRRC15 has been found to be significantly downregulated in OVX mice and upregulated upon osteogenic induction in a p65-dependent manner (*Wang et al., 2018*). Collagen XI, which encodes the $\alpha$1 chain of type XI collagen, is essential for collagen fibril formation in articular cartilage (*Blaschke et al., 2000*). Primary type II collagen with an alpha 1 collagen XI ($\alpha$1(XI)) chain structured into collagen fibrils forms a network in the cartilage ECM that contributes to the retention of proteoglycans and tensile strength in cartilage tissue (*Holyoak et al., 2018*). Many studies have shown that COL11A1 may increase genetic susceptibility to OA (*Wang et al., 2017a*; *Wang et al., 2017b*; *Styrkarsdottir et al., 2018*; *Fernández-Torres et al., 2020*). Changes in COL11A1 can affect chondrocyte maturation and collagen fiber thickness, potentially increasing the risk of various types of OA (*Jacobsen et al., 2024*). COL11A1 is in the same Wnt/$\beta$-catenin signaling pathway as GDF5 and has been identified as a candidate gene for OA (*Fernández-Torres et al., 2018*).

Inflammation and angiogenesis can alter the process of subchondral bone modeling (*Weber, Chan & Wen, 2019*). Immune infiltration analyses revealed that the infiltration of ly endothelial cells, mv endothelial cells, and endothelial cells was significantly positively correlated. There was also a positive correlation between CD4+ T cells and CD4+ Tems. Fewer adipocytes and endothelial cells infiltrated lesioned *versus* preserved samples, while MSCs, myocytes, plasma cells, Th2 cells, ly endothelial cells, and mv endothelial cells infiltrated more in lesioned samples.

## CONCLUSION

In conclusion, our findings indicate that the subchondral bone plays important roles in OA progression. Understanding the mechanisms by which subchondral bone is involved in OA

development and progression will greatly improve the diagnosis, treatment, and prognosis of OA patients. However, further investigations are necessary to validate the correlation between the key genes identified in this study and immune infiltration and elucidate the roles of these factors in OA progression.

## ACKNOWLEDGEMENTS

We thank LetPub for its linguistic assistance during the preparation of this manuscript.

### Funding

This study was supported by the National Clinical Research Center for Orthopedics, Sports Medicine & Rehabilitation (grant number: 2021-NCRC-CXJJ-PY-20), the Supported Project for "New Era Long Jiang Excellent Master's and Doctoral Thesis" (grant no. LJYXL2022-071), and the Dual Professor Cooperation Program "Diagnosis of Knee Osteoarthritis Based on Deep Learning Measurement of Cartilage Thickness" from the Second Affiliated Hospital of Harbin Medical University. The funders had no role in study design, data collection and analysis, decision to publish, or preparation of the manuscript.

### Grant Disclosures

The following grant information was disclosed by the authors:
The National Clinical Research Center for Orthopedics, Sports Medicine & Rehabilitation: 2021-NCRC-CXJJ-PY-20.
The Dual Professor Cooperation Program "Diagnosis of Knee Osteoarthritis Based on Deep Learning Measurement of Cartilage Thickness" from the Second Affiliated Hospital of Harbin Medical University.
The Supported Project for "New Era Long Jiang Excellent Master's and Doctoral Thesis": LJYXL2022-071.

### Competing Interests

The authors declare there are no competing interests.

### Author Contributions

- Gang Zhang performed the experiments, authored or reviewed drafts of the article, and approved the final draft.
- Jinwei Qin performed the experiments, authored or reviewed drafts of the article, and approved the final draft.
- Wenbo Xu performed the experiments, prepared figures and/or tables, and approved the final draft.
- Meina Liu analyzed the data, authored or reviewed drafts of the article, and approved the final draft.
- Rilige Wu conceived and designed the experiments, analyzed the data, prepared figures and/or tables, and approved the final draft.

- Yong Qin conceived and designed the experiments, authored or reviewed drafts of the article, and approved the final draft.

## Human Ethics

The following information was supplied relating to ethical approvals (*i.e.*, approving body and any reference numbers):

This study was approved by the Ethics Committee of the second Affiliated Hospital of Harbin Medical University (Approval number: ky2020-078) and informed consent was taken from all the patients.

## Data Availability

The raw data are available in the Supplemental Files.

## Supplemental Information

Supplemental information for this article can be found online at http://dx.doi.org/10.7717/peerj.17417#supplemental-information.

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
