# Peer review of "Gene expression and immune infiltration analysis comparing lesioned and preserved subchondral bone in osteoarthritis"

_PeerJ, doi:10.7717/peerj.17417_

## Round 0.1 · original submission · Major Revisions

· Academic Editor

Major Revisions

Please address the concerns of both reviewers and amend the manuscript accordingly.

**Language Note:** The review process has identified that the English language must be improved. PeerJ can provide language editing services - please contact us at [email protected] for pricing (be sure to provide your manuscript number and title). Alternatively, you should make your own arrangements to improve the language quality and provide details in your response letter. – PeerJ Staff

Reviewer 1 ·

Basic reporting

In this manuscript, Zhang et al. performed DEG analysis on a dataset (consist of 10 normal samples and 40 OA samples) obtained from GEO. The results were further analyzed with a series of bioinformatic approaches including functional annotation, protein-protein interaction network construction, xCell, etc.
This study reported over 1000 genes that were either up- or down- regulated in OA patients, many of which were involved the progression of OA. As a result, this manuscript would potentially provide guidance for future pathological and pharmaceutical studies.
However, this manuscript needs to be better formatted.
1. The resolution of all the figures are not high enough to see the details.
2. The figures should be better labeled. For example, in Fig.2A, the genes of interest should be high-lighted and labeled, otherwise the reader cannot get any information out of the volcano plot. In addition, it would be helpful if the authors could include more information in the figure legends.
3. In the main text, all the figure references are wrong (I guess Fig.1 was added after the text was written).
4. In lane 124-125, the authors claimed that LEP expression was increased in OA subchondral osteoblasts, but their analysis showed that LEP was one of the most down-regulated gene. Please clarify.
5. A few typos like in lane 28, [“Proteoglycans in cancer,”] should be [“Proteoglycans in cancer”.]. In lane 40, [It’s] should be [Its]. In lane 139, [upregulated] should be [upregulation].

Experimental design

The overall work flow applied in this study was standard.
1. Please explain why GSE51588 was picked for DEG analysis. Please also explain why rat dataset GSE30322 was included in the multi-set analysis for co-expression screening, while all the other datasets were from human.
2. Need to indicate how many repeats were done for qRT-PCR (Fig.7).

Validity of the findings

In this manuscript, the authors mostly focused on bioinformatic analysis. The experimental validation of results appeared to be limited.
The authors discussed on a lot of genes. I understand this was because the DEG analysis identified many genes. But it would be nice if the authors could pick a few genes that they found most interesting, and either further validate by western blot (or other techniques), or do more in-depth discussion in the “Discussion and Conclusion” part.

Reviewer 2 ·

Basic reporting

The manuscript by Zhang et al. is aimed at understanding the difference in gene expression and immune infiltration between lesioned and preserved subchondral bone in osteoarthritis. Overall, the study successfully identifies the genes and the proteins that can play a role in osteoarthritis using a combination of bioinformatics tools and qRT-PCR. Particularly, the authors should be commended for the Discussion section, which is well-written, and the information correlated with the latest studies. However, there are several outstanding issues regarding the use of the English language and representation of figures. Moreover, certain conclusions drawn by the authors do not support the data, and I have detailed these issues in the subsequent parts of this review.

1. The clarity of the text can be enhanced through improvements in the English language for a global readership. The manuscript consists of phrasing that hinders comprehension. I would suggest consulting a colleague well versed in English and informed regarding the subject matter. Alternatively, the authors can also consult a professional editing service.

2. The introduction needs more background. Specifically, lines 40-46 should be expanded to include more information about the pathology. Moreover, additional information is needed about the nature and role of alendronate. Similarly, more details are needed on lines 49-50 about the mentioned gene-level changes. The introduction should include a summary of the results and their implications.

3. In line 58, the acronym MT-LT has been introduced without a previous explanation.

4. Along with the title for all the figures, a summary of the findings gleaned from them would benefit the reader.

5. The figures are misnumbered in the results section. For example, in line 67, figure 2A refers to 3A.

6. The X-axis label of Figure 2B is unclear. Similarly, the inset graph’s size of the figure needs to be increased.

7. The font size of “Gene Number” and “log10” in Figures 3 and 4 needs to be increased. The values for fold-enrichment on the X-axis are unclear, and the font needs to be increased. The size of the graphs should be the same for each figure.

8. The font size for the DEGs in Fig 4A needs to be increased.

9. Fig. 6 would benefit from a legend explaining the color codes and the numbers in the Venn diagram.

10. Individual data points need to be shown in the bar graph of Figure 7. The size of the graphs should be the same, and they should not overlap with each other. Also, it is unclear why authors have MT, LT, and IL-1β in the X-axis, and a brief explanation of the findings in the legend would be helpful.

11. The font size for all the axes needs to be increased in Figure 8. The overlap between texts in the axis should be removed.

12. In the discussion section, a paragraph needs to be included in the beginning that briefly describes the premise of the study.

13. In the reference section, the authors need to mention the source for reference #26.

14. Table 1 and certain raw data files contain letters that are not in English. Moreover, in Table 1, the species names are not italicized.

15. In multiple lines of the manuscript, CD4+ T cells have been referred to as CD4 T+ cells, which need to be corrected. Additionally, there are several typos throughout the manuscript (such as CD4+ Tem cells mentioned as CD4 +Tem cells, improper capitalization of words and word spacing, etc.), which require attention.

16. The Abstract includes abbreviations not spelled out at first use.

17. Certain sections of the manuscript need rearrangement. Please follow the Research manuscript template under the manuscript guidelines for authors.

Experimental design

The study demonstrates a scientifically sound and largely well-executed experimental design. The materials and methods are clearly articulated, providing a comprehensible framework for potential replications. Nevertheless, I have outlined both minor and major suggestions for your consideration below.

1. For each of the subsections in the results section, the authors need to include a statement at the beginning addressing the purpose of the particular experiment/analysis described in that subsection.

2. For future ease of replication, lines 206-207 would benefit from having the concentration of collagenase and the temperature at which the experiments were conducted. Additionally, cell culture conditions also need to be reported. Similarly, reverse transcription reaction conditions such as the concentration of enzyme, the total amount of RNA, the temperature of the reaction, etc., need to be reported. Finally, qRT-PCR conditions, such as the amount of template DNA, primers, and polymerase concentration and its type, temperature, number of cycles, and other conditions, need to be detailed.

Validity of the findings

The conclusions do not always match the figures or outcomes of the experiments. A few instances have been listed below:

1. The authors do not present any supporting figure to show the most upregulated and downregulated genes, as mentioned in line 59. If the information is present in the supplementary, it needs to be made sufficiently easy to comprehend.

2. In line 127, the statement “And further research needs to be performed.” would benefit from further clarification about exactly what needs to be researched further.

3. In the second panel of Fig. 7, the difference in COL11A1 expression is similar in Chondroblast and not significantly different in Fibroblast-like synovial. Thus, the statement in lines 90-91 that the expression of COL11A1 is upregulated in all tissues is not supported by observed data.

---

## Round 0.2 · Minor Revisions

· Academic Editor

Minor Revisions

Please address the remaining issues pointed out by the reviewer and amend the manuscript accordingly.

Staff note: TIF is not supported by PeerJ. Please use PNG.

Reviewer 1 ·

Basic reporting

Most of my concerns were addressed in the revision.
It is appreciated that the resolution of figures was improved. In the final submission, it’s recommended to use .tif format, and make sure not to change the length:width ratio, so that the letters in the figures would not look strange.

Experimental design

My concerns were addressed in the revision.
It would be better if a brief explanation for the selection of the datasets studied in this manuscript is included in the main text.

Validity of the findings

All of my concerns were addressed in the revision.

Additional comments

No comments.

---

## Round 0.3 · accepted · Accept

· Academic Editor

Accept

The revied manuscript is acceptable now, as all the remaining issues pointed by the reviewer were adequately addressed.